# Motion Style Transfer: Modular Low-Rank Adaptation for Deep Motion Forecasting

**Parth Kothari**     **Danya Li**     **Yuejiang Liu**     **Alexandre Alahi**

École Polytechnique Fédérale de Lausanne (EPFL)

**Abstract:** Deep motion forecasting models have achieved great success when trained on a massive amount of data. Yet, they often perform poorly when training data is limited. To address this challenge, we propose a transfer learning approach for efficiently adapting pre-trained forecasting models to new domains, such as unseen agent types and scene contexts. Unlike the conventional fine-tuning approach that updates the whole encoder, our main idea is to reduce the amount of tunable parameters that can precisely account for the target domain-specific motion style. To this end, we introduce two components that exploit our prior knowledge of motion style shifts: (i) a low-rank motion style adapter that projects and adjusts the style features at a low-dimensional bottleneck; and (ii) a modular adapter strategy that disentangles the features of scene context and motion history to facilitate a fine-grained choice of adaptation layers. Through extensive experimentation, we show that our proposed adapter design, coined MoSA, outperforms prior methods on several forecasting benchmarks. Code available at https://github.com/vita-epfl/motion-style-transfer

**Keywords:** Motion Forecasting, Distribution Shifts, Transfer Learning

## 1 Introduction

Motion forecasting is an essential pillar for the successful deployment of autonomous systems in environments comprising various heterogeneous agents. It presents the challenges of modeling (i) universal etiquette (*e.g.*, goal-directed behaviors, avoiding collisions) that govern general motion dynamics of all agents; and (ii) social norms (*e.g.*, the minimum separation distance, preferred speed) that influence the navigation styles of different agents across different locations. Owing to the success of deep neural networks on large-scale datasets, learning prediction models in a data-driven manner has become a de-facto approach for motion forecasting and has shown impressive results [1, 2, 3, 4].

However, existing deep forecasting models suffer from inferior performance when they encounter novel scenarios [5, 6, 7, 8]. For instance, a network trained with large-scale data for pedestrian forecasting struggles to directly generalize to cyclists. Some recent methods propose to incorporate strong priors robust to the underlying distribution shifts [9, 10, 11]. Yet, these priors often make strong assumptions on the distribution shifts, which may not hold in practice. This shortcoming motivates the following transfer learning paradigm: adapting a forecasting model pretrained on one domain with sufficient data to new domains such as unseen agent types and scene contexts *as efficiently as possible*.

One common transfer learning approach is fine-tuning a pretrained model on the data collected from target domain. However, directly updating the model is often sample inefficient, as it fails to exploit the inherent structure of the distributional shifts in the motion context. In the forecasting setup, the physical laws behind motion dynamics are generally invariant across geographical locations and agent types: all agents move towards their goal and avoid collisions. As a result, the distribution shift can be largely attributed to the changes in the motion style, defined as the way an agent interacts with its surroundings. Given this decoupling of motion dynamics, it can be efficient for an adaptation algorithm to only account for the updates in the target motion style.

In this work, we efficiently adapt a deep forecasting model from one motion style to another. We refer to this task as *motion style transfer*. We retain the domain-invariant dynamics by freezing the

6th Conference on Robot Learning (CoRL 2022), Auckland, New Zealand.

pre-trained network weights. To learn the underlying shifts in style during adaptation, we introduce motion style adapters (MoSA), which are new modules inserted in parallel to the encoder layers. The style shift learned by MoSA is injected into the frozen pre-trained model. We hypothesize that the style shifts across forecasting domains often reside in a low-dimensional space. To formulate this intuition, we design MoSA as a low-dimensional bottleneck, inspired by recent works in language [12, 13]. Specifically, MoSA comprises two trainable matrices with a low rank. The first matrix is responsible for extracting the style factors to be updated, while the second enforces the updates. MoSA learns the style updates by adding and updating less than $2\%$ of the parameters in *each layer*.

In low-resource settings, it can be difficult for MoSA to distinguish the relevant encoder layers updates from the irrelevant ones, resulting in sub-optimal performance. To facilitate an informed choice of adaptation layers, we propose a modularized adaptation strategy. Specifically, we consider forecasting architectures that disentangle the fine-grained scene context and past agent motion using two independent low-level encoders. This design allows the flexible injection of MoSA to one encoder while leaving the other unchanged. Given the style transfer setup, our modular adaptation strategy yields substantial performance gains in the low-data regime.

We empirically demonstrate the efficiency of MoSA on the state-of-the-art model Y-Net [2] on the heterogenous SDD [14] and inD [15] datasets in various style transfer setups. Next, we highlight the potential of our modularized adaptation strategy on two setups: Agent Motion Style Transfer and Scene Style Transfer. Finally, to showcase the generalizability of MoSA in self-driving applications, we adapt a large-scale model trained on one part of the city to an unseen part, on the Level 5 Dataset [16]. Through extensive experimentation, we quantitatively and qualitatively show that given just 10-30 samples in the new domain, MoSA improves the generalization error by $25\%$ on SDD and inD. Moreover, our design outperforms standard fine-tuning techniques by $20\%$ on the Level 5 dataset.

## 2   Related Work

**Motion forecasting.** Classical models described the interactions between various agents based on domain knowledge but often failed to model complex social interactions in crowds [17, 18, 19]. Following the success of Social LSTM [1], various data-driven forecasting models have been proposed to capture social interactions directly from observed data [3, 20, 21, 4, 22, 23]. These methods heavily rely on a large and diverse set of training data, which may not be readily available for novel agents and locations. In this work, we efficiently adapt a pretrained model to unseen target domains.

**Distribution shifts.** The primary challenge in adapting to new domains lies in tackling the underlying distributional shifts. In the motion context, negative data augmentation techniques have been applied in a limited scope to reduce collisions [10] and off-road predictions [24] on new domains. Closely related to our work, Liu *et al.* [25] proposed to reuse the majority of pre-trained parameters for efficient adaptation. However, there exist key differences in the methodology: (1) we does not require access to multiple training domains with varying styles in order to perform adaptation (2) we introduce low-rank adapters instead of finetuning existing parameters, to model domain shifts. Domain adaptation is another paradigm that allows a learning algorithm to observe unlabelled test samples. While this approach is effective for supervised visual tasks [26, 27, 28], it is not ideal for motion forecasting where the crucial challenge is sample efficiency as labels (future trajectories) are fairly easy to acquire. Therefore, we propose to perform transfer learning using limited data.

**Transfer learning.** The standard approach of fine-tuning the entire or part of the network [29, 30] has been shown to outperform feature-based transfer strategy [31, 32]. In the motion context, transfer learning given limited data often requires special architecture designs like external memory [33] and meta-learning objectives [34, 35] that require access to multiple training environments. Wang *et al.* [36] performed online adaptation across different scenarios for vehicle prediction domains. Recently, there has been a growing interest in developing parameter-efficient fine-tuning (PET) methods in both language and vision, as they yield a compact model [37, 12, 13] and show promising results in outperforming fine-tuning in low-resource settings [13, 38]. Similar in spirit to PET methods, we introduce additional parameters in our network that account for the updates in target style.

**Motion Style.** the popular work of Robicquet et al. [14] defined *navigation style* as the way different agents interact with their surroundings. It introduced social sensitivity as two handcrafted descriptions of agent style and provided them as input to the social force model [17]. In this work, we model style

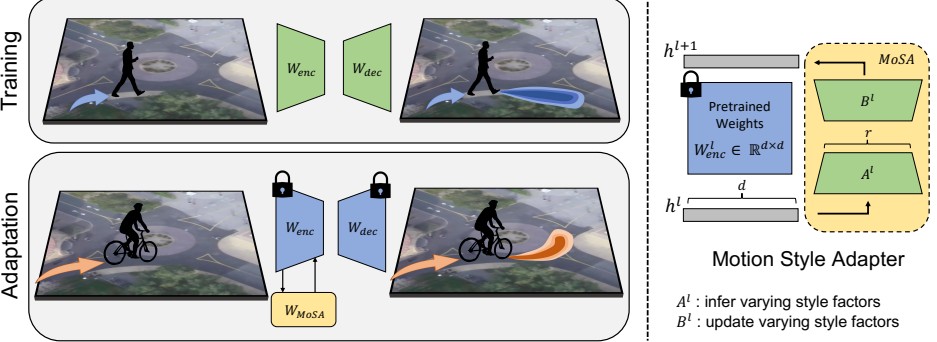

Figure 1: We present an efficient transfer learning technique that adapts a forecasting model trained with sufficient labeled data (*e.g.*, pedestrians), to novel domains exhibiting different motion styles (*e.g.*, cyclists). We freeze the pretrained model and only tune a few additional parameters that aim to learn the underlying style shifts (left). We hypothesize that the style updates across domains lie in a low-dimensional space. Therefore, we propose motion style adapters with a low-rank decomposition ($r \ll d$), designed to infer and update the few style factors that vary in the target domain (right).

as a latent variable that is learned in a data-driven manner. Furthermore, we decouple motion style into scene-style components and agent-style components to favour efficient adaptation.

## 3 Method

Deep neural networks have shown remarkable performance in motion forecasting thanks to the availability of large-scale datasets. However, these models often struggle to generalize when unseen scenarios are encountered in the real world due to underlying differences in motion style. In this section, we first formally introduce the problem setting of motion style transfer. Subsequently, we introduce the design of our style adapter modules that help to effectively tackle motion style transfer. Finally, we describe the application of style adapters to a modularized architecture in order to perform style transfer in a more efficient manner.

### 3.1 Motion Style Transfer

**Motion style.** Modelling agent motion behavior involves learning the social norms (*e.g.*, minimum separation distance to others, preferred speed, valid areas of traversal) that dictate the motion of the agent in its surroundings. These norms differ across agents as well as locations. For instance, the preferred speed of pedestrians differs from that of cyclists; the separation distance between pedestrians in parks differs from that in train stations. To describe these agent-specific (or scene-specific) elements that govern underlying motion behavior, we define the notion of "motion style". Motion style is the collective umbrella that models the social norms of an agent given its surroundings.

**Problem statement.** Consider the inductive transfer learning setting for deep motion forecasting across different motion styles. Specifically, we are provided a forecasting model trained on large quantities of data comprising a particular set of style(s) and our goal is to adapt the model to the idiosyncrasies of a target style as efficiently as possible. We denote the model input and ground-truth future trajectory of an agent $i$ using $x_i$ and $y_i$ respectively. The input $x_i$ comprises the past trajectory of the agent, surrounding neighbors, and the surrounding context map. The context can take various forms like RGB images or rasterized maps. We assume that the data corresponding to an agent type is generated by an underlying distribution $\mathcal{P}_{X,Y}(\cdot; s)$ parameterized by $s$, the style of the agent. As mentioned earlier, the style is dictated by both the agent type and its surroundings.

**Training.** The forecasting model has an encoder-decoder architecture (see Fig. 1) with weights $W_{enc}$ and $W_{dec}$ respectively. The training dataset, $D_S$ of size $N$ is given by $\cup_{s \in S} D_s = (x_i, y_i)_{i \in \{1, \dots, N\}}$, where $S$ is a collection of motion styles observed within the dataset. The model is trained to minimize a loss objective $\mathcal{L}$, such as the negative log likelihood (NLL) [1, 3] loss or variety loss [39]:

$$\mathcal{L}_{train}(D_S; W_{enc}, W_{dec}) = \frac{1}{N} \sum_{i=1}^{N} \mathcal{L}(x_i, y_i; W_{enc}, W_{dec}). \quad (1)$$

**Adaptation.** When a novel scenario with style $s'$ ($s' \notin S$) is encountered, it leads to a distribution shift and the learned model often struggles to directly generalize to the corresponding dataset $D_{s'} = (x'_i, y'_i)_{i \in \{1, \ldots, N_{target}\}}$ of size $N_{target}$. The common approach to tackling such shifts is to fine-tune the entire or part of the pretrained model. Fine-tuning optimizes an objective similar to training, but on the new dataset:

$$\mathcal{L}_{adapt}(D_{s'}; W_{enc}, W_{dec}) = \frac{1}{N_{target}} \sum_{i=1}^{N_{target}} \mathcal{L}(x'_i, y'_i; W_{enc}, W_{dec}). \qquad (2)$$

In this work, we aim to develop an adaptation strategy for efficient motion style transfer, *i.e.*, cases where $N_{target}$ is small ($N_{target} \ll N$). Often, motion behaviors do not change drastically across domains. Instead, most of the behavioral dynamics are governed by universal physical laws (*e.g.*, influence of inertia, collision-avoidance). We therefore propose to freeze the weights of the pretrained forecasting model and introduce motion style adapters, termed *MoSA*, to capture the target motion style. As shown Fig. 1, we adapt a pre-trained forecasting model by fine-tuning $W_{MoSA}$ with the following objective:

$$\mathcal{L}_{adapt}(D_{s'}; W_{MoSA}) = \frac{1}{N_{target}} \sum_{i=1}^{N_{target}} \mathcal{L}(x'_i, y'_i; W_{MoSA}). \qquad (3)$$

## 3.2 Motion Style Adapters

Our main intuition is that the style shifts across forecasting domains are usually localized – they are often due to the changes in only a few variables of the underlying motion generation process. Therefore, during style transfer, we only need to adapt the distribution of this small portion of latent factors, while keeping the rest of the factors constant. These updates would correspond to the changes in motion style ($s \rightarrow s'$) in the target domain, as the general principles of motion dynamics (*e.g.*, avoid collisions, move towards goal) remain the same across domains for all agents. We design motion style adapters, referred to as MoSA, to carry out these updates.

Our proposed MoSA design comprises a small number of extra parameters added to the model during adaptation (see Fig. 1). Each module comprises two trainable weight matrices of low rank, denoted by $A$ and $B$. The first matrix $A$ is responsible for inferring the style factors that are required to be updated to match the target style, while the second matrix $B$ performs the desired update. The low rank $r$ realizes our intuition that style updates reside in a low-dimensional space, by restricting the number of style factors that get updated ($r \ll d$ where $d$ is the dimension size of an encoder layer). Therefore, during adaptation, the weight updates of the encoder are constrained with our low-rank decomposition $W_{MoSA} = BA$. The pretrained model is frozen and only $A$ and $B$ are trained.

For brevity, let us consider the adaptation of encoder layer $l$ with input $h^l$ and output $h^{l+1}$. As shown in Fig. 1, $W_{enc}^l$ and $W_{MoSA}^l$ are multiplied with the same input $h^l$, and their respective output vectors are summed coordinate-wise as shown below:

$$\text{(Train)} \quad h^{l+1} = W_{enc}^l h^l, \qquad (4)$$

$$\text{(Adapt)} \quad h^{l+1} = W_{enc}^l h^l + W_{MoSA}^l h^l = W_{enc}^l h^l + B^l A^l h^l. \qquad (5)$$

It has been shown that initialization plays a crucial role in parameter-efficient transfer learning [37, 12]. Therefore, following common practices, matrices $A$ and $B$ are initialized with a near-zero function [12], so that the original network is unaffected when training starts. Furthermore, the initialization provides flexibility to these modules to ignore certain layers during motion style updates. Despite this flexibility, the total number of extra parameters is significant and can be inconducive to efficient style transfer. Therefore, to further boost sample efficiency, we present a modular adaptation strategy which we describe next.

## 3.3 Decoupling Motion Style Adapters

Motion style can be decoupled into scene-specific style and agent-specific style. Scene-specific style dictates the changes in motion due to physical scene structures. For instance, cyclists prefer to stay on the lanes while pedestrians move along sidewalks. The agent-specific style captures the underlying

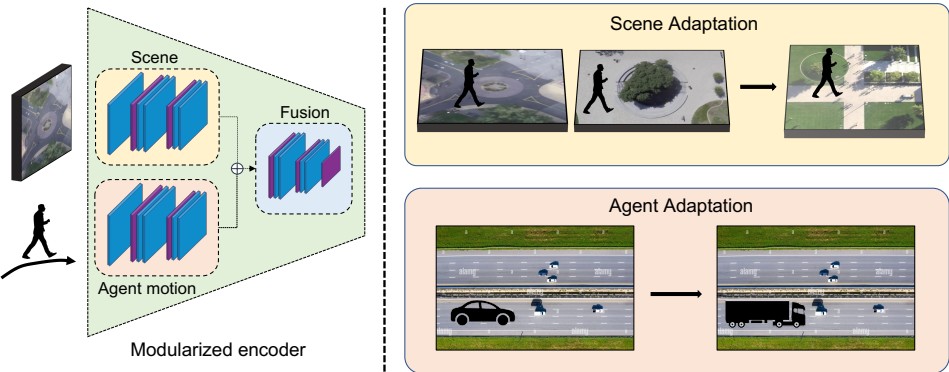

Figure 2: Our modular style transfer strategy updates only a subset of the encoder to account for the underlying style shifts. For instance, we adapt the scene encoder only to model scene style shifts (top right). While for the underlying agent motion shift, we only update the agent motion encoder (bottom right). This strategy boosts performance in low-resource settings.

navigation preferences of different agents like distance to others and preferred speed. Modularizing the encoder into two parts that account for the physical scene and agent's past motion independently can help to decouple the functionality of our motion style adapters and further improve performance.

Consider the modularized motion encoder designs shown in Fig. 2. The modularized encoder models the input scene and agent's past history independently. The fusion encoder then fuses the two representations together. This design has the advantage to decouple the task of the style adapters into scene-specific updates and agent-specific updates. Given the modularized setup, the nature of the underlying distribution shifts can help guide which modules within the model are required to be updated to the target style. As demonstrated in Section. 4.4, given different categories of style transfer setups, decoupling style adapters can improve the adaptation performance while significantly reducing the number of updated parameters.

## 4 Experiments

We evaluate our method on a total of three datasets to study the performance of motion style adapters: Stanford Drone Dataset (SDD) [14], the Intersection Drone Dataset (InD) [15], and Level 5 Dataset (L5) [16]. We evaluate each method over five experiments with different random seeds. More implementation details and ablations are summarized in the Appendix.

**Baselines.** We use the state-of-the-art Y-Net model [40] on SDD and inD, and the Vision Transformer (ViT) [41] on L5 across all methods. We compare the following:

- Full Model Finetuning (FT) [29]: we update the weights of the entire model.
- Partial Model Finetuning (ET) [25]: we update the weights of the Y-Net encoder for SDD and inD , and the last two layers of ViT for Level 5.
- Parallel Adapters (PA) [42]: we insert a convolutional layer with a fixed filter size in parallel to each encoder layer and update the weights of these added layers. This baseline does not incorporate the low-rank constraint.
- Adaptive Layer Normalization [43, 44]: we update the weights and biases of the layer normalization.
- Motion Style Adapters (MoSA) [Ours]: we insert our motion style adapters in parallel to each encoder layer in SDD and inD, and in parallel to query and value matrices of multi-headed attention in Level 5. During modularized adaptation, we add our modules only across the specified encoders.

**Metrics.** We use the established Average Displacement Error (ADE) and Final Displacement Error (FDE) metrics for measuring the performance of model predictions. ADE is calculated as the $l_2$ error between the predicted future and the ground truth averaged over the entire trajectory while FDE is the $l_2$ error between the predicted future and ground truth for the final predicted point [1]. For multiple predictions, the final error is reported as the `min` error over all predictions [39]. Additionally, we define the generalization error as the error of the pretrained model on the target domain. The more the dissimilarity between the source domain and target domain, the higher the generalization error.

Table 1: Evaluation of adaptation methods for motion style transfer (pedestrians to cyclists) on SDD and scene style transfer on InD using few samples $N_{target} = \{10, 20, 30\}$. Error reported is Top-20 FDE in pixels. The generalization error on SDD is 58 pixels and on inD is 33 pixels. Our proposed motion style adapters (MoSA) outperform competitive baselines and improve upon the generalization error by $> 25\%$ in both setups. Mean and standard deviation were calculated over 5 runs.

| | Stanford Drone Dataset | | | Intersection Drone Dataset | | |
|---|---|---|---|---|---|---|
| $N_{target}$ | 10 | 20 | 30 | 10 | 20 | 30 |
| FT | $57.28 \pm 1.21$ | $52.61 \pm 0.87$ | $46.31 \pm 1.79$ | $27.92 \pm 1.99$ | $25.15 \pm 1.08$ | $23.18 \pm 0.64$ |
| ET [25] | $51.88 \pm 1.32$ | $46.78 \pm 1.78$ | $43.13 \pm 1.03$ | $28.06 \pm 0.68$ | $23.19 \pm 1.39$ | $21.13 \pm 1.00$ |
| PA [42] | $52.77 \pm 0.85$ | $47.75 \pm 1.83$ | $44.70 \pm 1.28$ | $28.71 \pm 1.50$ | $26.10 \pm 0.74$ | $25.00 \pm 1.08$ |
| MoSA (ours) | $\mathbf{49.98 \pm 1.05}$ | $\mathbf{45.55 \pm 0.77}$ | $\mathbf{41.69 \pm 0.88}$ | $\mathbf{25.18 \pm 0.72}$ | $\mathbf{21.70 \pm 0.84}$ | $\mathbf{20.35 \pm 1.18}$ |

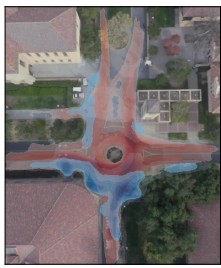

Figure 3: Heatmap of pedestrians motion (in blue) and cyclists motion (in red) on SDD *deathCircle* location.

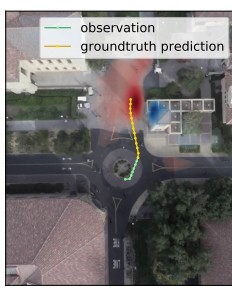 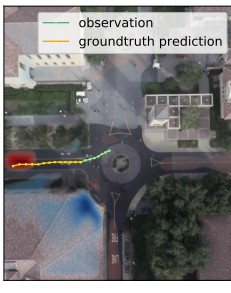 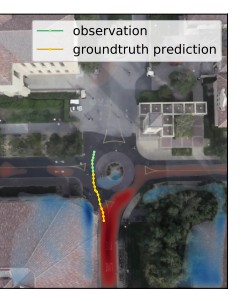

Figure 4: Illustration of the difference in goal decoder output of Y-Net, pre-trained on pedestrians, after model adaptation on cyclist data using our motion style adapters. Using only 20 samples, Y-Net learns to focus on the road lanes (red) instead of sidewalks (blue) for cyclist forecasting.

## 4.1 Motion Style Transfer across Agents on Stanford Drone Dataset

We perform short-term prediction, where the trajectory is predicted for the next 4.8 seconds, given 3.2 seconds of observation. The Y-Net model is trained on pedestrian data across *all* scenes and adapted to the cyclists' data in *deathCircle_0*, as there exists a clear distinction between the motion style of pedestrians and cyclists (see Fig. 3). We adapt the model using $N_{target} = \{10, 20, 30\}$ samples.

Tab. 1 quantifies the performance of various adaptation techniques. The model trained on pedestrians does not generalize to cyclists as evidenced by the high generalization error of 58 pixels. Our MoSA design reduces this error by $\sim 30\%$ using only 30 samples. Moreover, MoSA outperforms the baselines while keeping the pretrained model frozen and updating only $2\%$ additional parameters. Fig. 4 illustrates the updates in the Y-Net goal decoder output (red means increase in focus and blue means decrease in focus) on model adaptation using MoSA. Adapted Y-Net successfully learns the style differences between the behavior of pedestrians and cyclists: 1) it correctly infers valid areas of traversal, 2) effectively captures the multimodality of cyclists, and 3) updates the motion style parameters as the new cyclist goal positions (red) are farther from the end of the observation position, compared to the un-adapted goal positions (blue).

## 4.2 Motion Style Transfer across Scenes on Intersection Drone Dataset

We perform long-term prediction, where trajectory in the next 30 seconds is predicted, given 5 seconds of observation. The Y-Net model is trained on pedestrians in $\{scene2, scene3, scene4\}$ and tested on unseen scene $scene1$. We adapt the model using $N_{target} = \{10, 20, 30\}$ samples.

Despite the long-term prediction setup, the generalization error is 33 pixels which is lower compared to SDD, as the target domain is more similar to the source domain. Tab. 1 quantifies the performance of scene style transfer across all methods. Using just 30 samples, MoSA improves the generalization error by $\sim 40\%$ and outperforms its counterparts. The superior performance in comparison to PA justifies the importance of the low-rank constraint.

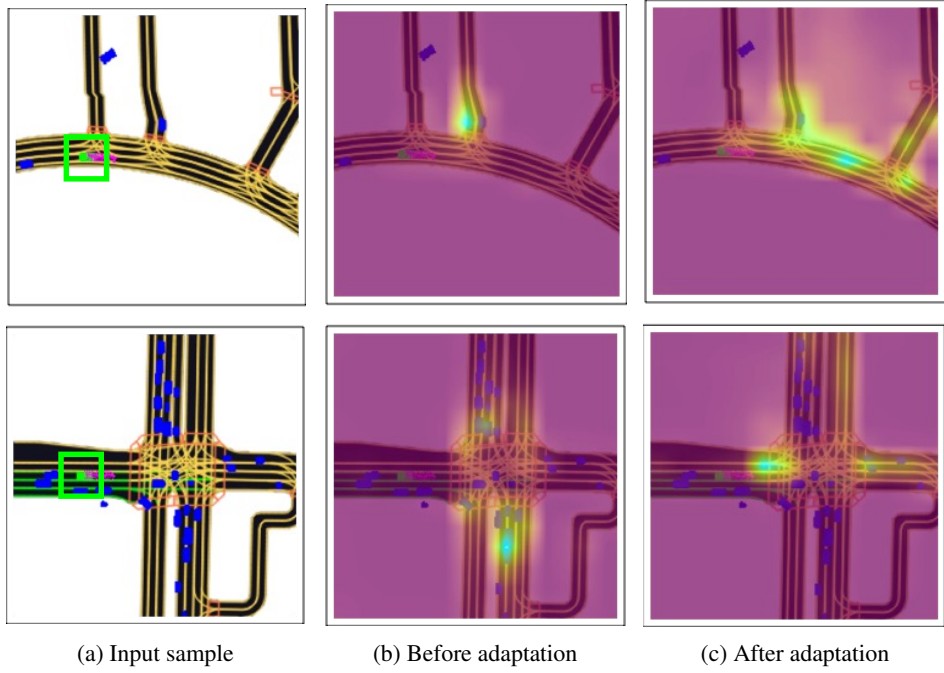

(a) Input sample        (b) Before adaptation        (c) After adaptation

Figure 5: Attention heatmaps of the last layer of ViT before and after model adaptation on unseen route in Level 5. After adaptation, the attention maps are more refined. The ego (in green box) better focuses on the different possible future routes and the vehicles in front.

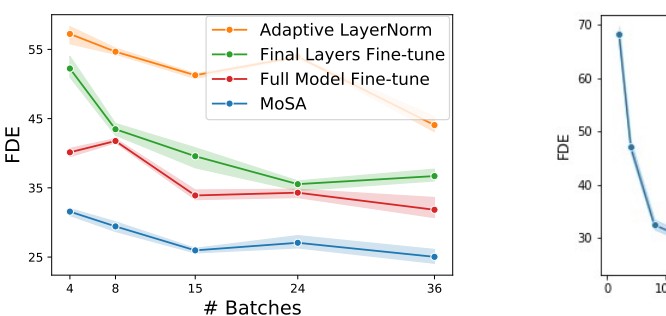

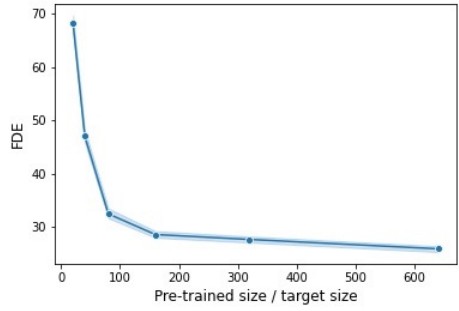

Figure 6: Evaluation of adaptation techniques for long-term motion prediction (25 secs) on Level 5. Error in meters for 5 seeds.

Figure 7: Impact of the pre-trained dataset size (relative size shown on x-axis) on motion adaptation on L5. Target dataset size fixed to 15 batches.

### 4.3 Motion Style Transfer across Scenes on Lyft Level 5 Dataset

We divide the L5 dataset into two splits based on the data collection locations thereby, constructing a scene style shift scenario. We train the ViT-Tiny model on the majority route and adapt it to the smaller route not seen during training. To simulate low-resource settings, we provide the frames, sampled at different rates, that cover the unseen route only once.

Fig. 6 quantitatively evaluates the performance of various adaptation strategies. MoSA performs superior in comparison to different baselines while adding and updating only $5\%$ of the full model parameters. It is apparent that our low-rank design is a smarter way of adapting models in the motion context as compared to fine-tuning the model. Fig. 7 empirically validates that a bigger pre-training dataset size results in better adaptation performance. Finally, Fig. 5 qualitatively illustrates the improvement of the attention heatmaps of the last layer of ViT post model adaptation using MoSA. MoSA helps to better focus on the different possible future routes and the vehicles in front. Additional visualizations are provided in the Appendix.

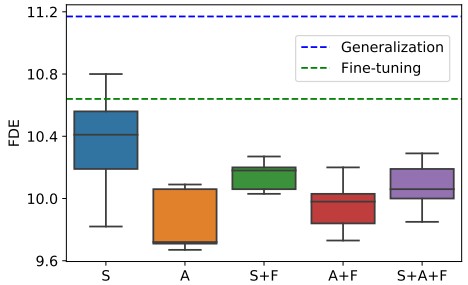 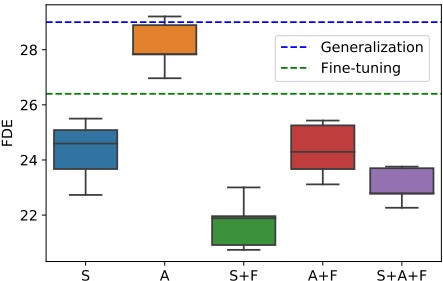

Figure 8: Agent Motion Transfer from *cars* to *trucks* on InD with $N_{target} = 20$ samples using different MoSA configurations. [A] performs best while [S] worsens performance. Error is Top-20 FDE in pix.

Figure 9: Scene Transfer on InD *pedestrians* with $N_{target} = 20$ samples using different MoSA configurations.[S+F] performs best while [A] worsens performance. Error is Top-20 FDE in pix.

### 4.4 Modular Motion Style Adapters

Now, we demonstrate the effectiveness of applying our adapters on top of a modularized architecture on two setups: Scene generalization and Agent Motion generalization. As shown in Fig. 2, we modularize the Y-Net architecture. We treat scene and agent motion independently for the first two layers of the encoder before fusing the learned representations. We refer to this design as Y-Net-Mod. Given Y-Net-Mod, we consider five cases based on modules on which MoSA is applied: (1) scene only [S], (2) agent motion encoder only [A], (3) scene and fusion encoder [S+F], (4) agent motion and fusion encoder [A+F], and (5) scene, agent motion and fusion encoders together [S+A+F].

**Agent motion generalization:** In $scene1$ of inD, cars and trucks share the same scene context differing only in their velocity distribution. Fig. 8 represents the performance of style transfer from cars to trucks on 20 samples under five different adaptation cases. It is interesting to note that adapting the agent motion encoder alone [A] performs the best while including the scene encoder for adaptation deteriorates performance ([S] worse than [A], [S+A+F] worse than [A+F]).

**Scene generalization:** We train the Y-Net-Mod model on pedestrian data on scene ids = $\{2, 3, 4\}$ and adapt it on $scene1$ of inD. Fig. 9 represents the performance of scene style transfer on 20 samples in the five cases. Contrary to the previous setup, adapting the scene encoder [S] is clearly more important than the agent motion encoder [A]. Further, adapting the agent encoder deteriorates performance ([S+A+F] worse than [S+F]). It is clear that modularization helps to boost the performance of MoSA.

## 5 Limitations

We demonstrated the effectiveness of decoupling our proposed motion style adapters using Y-Net-Mod. However, during training, we do not enforce any constraints on the learning objective to favor effective modularization that can result in more efficient adaptation. Developing training strategies that allow quick adaptation using MoSA is a potential line of future work. Another limitation is the requirement of human intervention in determining the modules to be adapted given the nature of the style shift. A future direction is automating the selection of layers to be adapted for target domains.

## 6 Conclusion

We tackle the task of efficient motion style transfer wherein we adapt a pre-trained forecasting model using limited samples from an unseen target domain. We hypothesize that the underlying shift across domains often resides in a low-dimensional space. We formulated this intuition into our motion style adapter (MoSA) design, which is trained to infer and update the style factors of variation in the target domain while keeping the pre-trained parameters frozen. Additionally, we present a modularized adaptation strategy that updates only a subset of the model given the style transfer setup. Extensive experimentation on three real-world datasets demonstrates the effectiveness of our approach.

**Acknowledgments**

This work was supported by Honda R&D Co. Ltd, the Swiss National Science Foundation under the Grant 2OOO21-L92326, and EPFL. We also thank VITA members and reviewers for their valuable comments.

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
