# OpenReview forum: "Motion Style Transfer: Modular Low-Rank Adaptation for Deep Motion Forecasting"
_robot-learning.org/CoRL/2022/Conference — CoRL 2022 Poster_

### Official Review · Reviewer_krte · 2022-07-29

**Originality:** Good
**Technical Quality:** Fair
**Clarity Of Presentation:** Good
**Impact:** 2

**Recommendation:**

Weak Accept: I recommend accepting the paper, but will not argue for my recommendation if the majority of other reviewers have a different opinion.

**Summary:**

The paper introduces a few-shot adaptation architecture for motion style transfer. The architecture simple uses two learnable matrices  (A and B) that are inserted at each layer and the output (B^l * A^l * h^l) is simply added, coordinate wise, to the original output of each hidden layer of the original network (whose weights are held frozen). The proposed technique has far-fewer trainable parameters than fine tuning the original network, for example. The paper claims that these parameters capture the "style" of motion or the essence of the new "scene". An additional decomposition is explored in which the scene and agent information are modularized and separate weights are depending depending on the labeled data (agent, scene, or both have changed). Comparisons to competing methods on three datasets show that the proposed method is effective at obtaining superior performance on standard error metrics.

**Issues:**

-There are several unsubstantiated, intuitive claims in the paper. For example: lines 142-143 and 165-168.
-Although I do not fully understand why, the method apparently requires "human intervention" (in the form of labels?) as outlined in 255-257.
-The authors mention that a comparison to methods that use "adaptive normalization layers" (e.g. FiLM, or [29]) would be presented, but I did not find this comparison in the experiments.

**Quality Of The Limitations Section:**

Additional details required

**Reviewer Expertise:**

3: The reviewer is fairly confident that the evaluation is correct

**Robotics Focus:**

Relevant but unlikely to deploy to hardware in near future

**Strengths And Weaknesses:**

Strengths:
-The proposed method is conceptually simple
-The proposed method appears to achieve superior results under apparently fair comparisons

Weaknesses:
-There are several unsubstantiated, intuitive claims in the paper.
-The method apparently requires "human intervention".
-Missing comparison to methods that use "adaptive normalization layers"

**Summary Of Recommendation:**

I recommend acceptance for two main reasons: 1-the proposed method seems to advance the state of the art on three datasets, 2-the proposed technique appears to be simple and fairly reproducible (*provided the mention of "human intervention" in the limitations section is is clarified).

---

> ### Author Response · Authors · 2022-08-20
> **Author Response to Reviewer krte**
>
> Thank you for the review. Below we address the raised concerns.
>
> > **Does the method require human intervention as outlined in L255-257.**
>
> - We’d like to clarify that our method does not require hard human interventions. We only leverage the human domain knowledge about the type of the underlying style shift to update the corresponding modules. i) Such high-level domain knowledge about the difference from one agent/environment to another is often widely available in practice. ii) By exploiting this domain knowledge through the modular low-rank adapter, our method substantially boosts the transfer learning efficiency. iii) This assumption can be further relaxed in the future, e.g., by measuring the distribution shifts in the feature space per module and automatically selecting the ones that vary the most.
>
> > **Comparison to “adaptive normalization layers”.**
>
> - We compare to the adaptive layer normalization baseline on the Level 5 dataset (blue plot in Fig 5), where we adapt the parameters of the layer normalization and the final linear layer of the vision transformer. Our motion style adapter performs superior to adaptive layer normalization. For the SDD/inD experiments, we utilize the state-of-the-art Y-Net model. The Y-Net architecture does not contain any normalization layers. Thus, we do not report the adaptive normalization baseline in that setup.
>
> > **Writing: L142-143 and L165-168 read as an unsubstantiated, intuitive claims in the paper.**
>
> - We agree that L142-143 reflects our intuition (and not evidence) behind the design and functionality of matrices A and B. The phrasing of L142-L143 will be modified accordingly in the final text.
>
> - Regarding L165-168, we will clarify that the two low-level encoders (and not the ‘modularized encoder’) independently model the input scene and agent’s history. As seen in Fig 2 (and Fig. 2 of the supplementary), the scene encoder has access to only the physical scene context while the agent motion encoder has access to only the past agent motion. Thus, these two encoder modules model the two input modalities independently and are decoupled.

---

> > ### Comment · Reviewer_krte · 2022-08-23
> > **Acknowledgment of Author Response to Review**
> >
> > Thank you for your clarification that the human intervention requirement described on L255-257 is not a serious constraint in practice, comparison to layer norm, and proposed minor modifications.
> > My ratings remain unchanged.

---

### Official Review · Reviewer_Qhox · 2022-08-01

**Originality:** Good
**Technical Quality:** Fair
**Clarity Of Presentation:** Good
**Impact:** 2

**Recommendation:**

Weak Reject: I recommend rejecting the paper, but will not argue for my recommendation if the majority of other reviewers have a different opinion.

**Summary:**

The paper is interested in motion forecasting, and specifically the problem of adapting a motion forecasting model to some change in "style", which is either a scene change or change in agent behavior. The key insight is that in the low-data regime, finetuning the whole model or some parts may be ineffective, and its better to train a new, low rank matrix to predict a residual to the frozen models prediction. Experiments on some standard datasets suggest this is better than full or partial finetuning. Additionally the paper proposes to explores decomposing the encoder into a separate "scene" and "agent" part.

**Issues:**

See Strengths and Weaknesses.

**Quality Of The Limitations Section:**

Limitations are addressed clearly

**Reviewer Expertise:**

2: The reviewer is willing to defend the evaluation, but it is quite likely that the reviewer did not understand central parts of the paper

**Robotics Focus:**

Relevant but unlikely to deploy to hardware in near future

**Strengths And Weaknesses:**

*Strengths*

- Overall it makes sense that motion forecasting models may need to be adapted to new scenarios, and low-data methods for adaptation are important.
- The approach of training a residual model that is constrained to be low rank makes sense as a way to adapt the model while optimizing a small number of parameters (which is important in the low-data regime).
- The method overall is pretty straightforward and explained in a clear way.

*Weaknesses/Questions*

Method:
- Overall I think the method makes sense for the low-data regime, though I'm not sure how novel the idea of adaptation by training a residual over a frozen model is. For example in robot learning many papers learn residual policies over an expert for adaptation [1], and in general in the idea of training a new layer on top of a frozen model is commonplace in machine learning. At the same time, I think applying these ideas to a new problem like motion forecasting is worthwhile.
- The low rank constraint on the residual layer is an interesting addition. Are the authors aware of other works in the meta-learning/finetuning literature that apply a similar constraint?
- I'm not entirely convinced by the claim of the decomposed encoder which separates style and movement components of motion prediction. As far I understand, nothing explicitly forces each head to focus on style/motion respectively correct? How can the model be guaranteed to decompose the input into those respective components?

Experiments:
- It would be worthwhile to include more ablations which highlight the benefits of the proposed method. For example, if it is indeed the case that the low-rank constraint on the residual weights is important, the paper should include ablations which remove and also vary the size of r, and analyze how it impacts performance.
- Additionally, for the experiment which looks at the decomposed Y-net into the scene and motion components, I don't see a head-to-head comparison between including vs. not including the decomposition. The experiment instead shows the impact of applying MoSA to the different parts of the decomposed module, but I don't see how the conclusion "It is clear that modularization helps to boost the performance of MoSA." can be drawn from Figure 7 and 8.

Fit:
- Overall, I think the focus of this paper is on effective adaptation of a pre-trained model using a residual layer. While it's applied to motion forecasting, the same MoSA method seems like it could be beneficial in general. So I wonder if this paper might be better evaluated at a learning/vision conference. This paper also does not apply the method on any real or simulated robotic control problem.

[1] Johannik et al. Residual Reinforcement Learning for Robot Control. 2018.

**Summary Of Recommendation:**

Overall proposes an interesting approach for adapting motion forecasting models that seems to improve prediction quality, but more thorough experiments are needed to support all of the claims about the method, and ultimately the work is not applied on any real or simulated robotic control problem.

---

> ### Author Response · Authors · 2022-08-20
> **Author Response to Reviewer Qhox (Part 2/2)**
>
>
> > **I don't see a head-to-head comparison between including vs. not including the decomposition, to support the conclusion "It is clear that modularization helps to boost the performance of MoSA.**
>
> - We would like to clarify that we claim that the ‘modular adaptation strategy’ (and not ‘modularization of encoder’) helps to boost the adaptation performance (see L10-12). Our main message in Sec 4.4 is that if the style transfer setup is known a-priori, we can determine the specific modules of the modularized encoder that need to be adapted to provide optimal performance. For instance, in the agent motion transfer scenario, updating the agent motion module [A] should improve performance while updating the scene module [S] should deteriorate performance. Fig 7 and Fig 8 validate our argument experimentally.
> - We will update the claim in the final text as “the modular adaptation strategy further helps to boost the performance of adaptation.” for clarity.
>
> > **In modularization, there is no explicit forcing for each head to focus on style/motion respectively.**
>
> - We would like to clarify that in modularization, each head is designed to focus on the physical scene context and past agent motion respectively (and not style/motion), As mentioned in L252-254, we agree that currently we do not enforce any additional constraint on the representations learned by the two low-level encoders during training. Developing training strategies that focus on explicitly decoupling the learned representations is an interesting avenue for future work. For instance, in Fig 8. currently, [S+F] performs better than [S]. A training algorithm that keeps the fusion encoder [F] fixed and only updates [S] across different training scenes, can potentially lead to improved adaptation performance of [S] compared to [S+F].
>
> > **Paper Fit and application to real or simulated robotic control problem.**
>
> - Motion prediction has been an integral part of the robotic autonomy stack. Contributions of many motion forecasting works [5-8] on real-world prediction problems have been recognized by CoRL. Thus, we believe that our work, with experiments on real-world forecasting datasets, can have a good impact on the performance of various robotic applications.
>
> [1] Johannik et al. Residual Reinforcement Learning for Robot Control. 2018.
>
> [2] Hu et al. Low-Rank Adaptation of Large Language Models, Arxiv 2021.
>
> [3] Mahabadi et al. COMPACTER: Efficient Low-Rank Hypercomplex Adapter Layers, NeurIPS 2021.
>
> [4] S.-A. Rebuffi, H. Bilen, and A. Vedaldi. Efficient parametrization of multi-domain deep neural networks. CVPR 2018.
>
> [5] Deo et al, Multimodal Trajectory Prediction Conditioned on Lane-Graph Traversals, CoRL 2021
>
> [6] Jia et al. Multi-Agent Trajectory Prediction by Combining Egocentric and Allocentric Views, CoRL 2021
>
> [7] Zhu et al. Motion Forecasting with Unlikelihood Training in Continuous Space, CoRL 2021
>
> [8] J. L. Houston et al. One thousand and one hours: Self-driving motion prediction dataset. In CoRL, 2020.

---

> > ### Comment · Reviewer_Qhox · 2022-08-28
> > **Re Author Response**
> >
> > Thanks to the authors for the detailed response.
> >
> > Regarding the ablations on the rank - thanks for adding these experiments, I think these are a good addition. It makes sense that too low of a rank hurts performance but too high also hurts performance.
> >
> > Regarding the modularization - I see, so the pre-trained models are already decomposed into a scene, motion, and fusion modules. And then you just selectively adapt different modules. Then the results make sense. Though I think that is probably expected, and I'm not sure I would consider it a major contribution of the paper.
> >
> > Regarding real robot results - I agree that motion forecasting is absolutely relevant to the robotics community. My concern was that all the results were simply offline evaluation on a static dataset. I think the claims of the paper would be strengthened if the method is used in the loop of a simulated/real robotic control problem, and led to less collisions.
> >
> > Overall, I think the paper is improved, and would raise my score to borderline accept.

---

> ### Author Response · Authors · 2022-08-20
> **Author Response to Reviewer Qhox (Part 1/2)**
>
> Thank you for the detailed review and thoughtful feedback. Below we answer specific questions.
>
> > **Novelty in addition to training a residual over the frozen model**
>
> - Indeed, training a residual has been shown effective in other domains like robotics [1]. Yet, it’s not clear what kind of residual is most effective for adaptation, and where they should be introduced. To address these open questions, we introduce two key components specific to motion forecasting that constitute our contributions: (i) we model the residual as a low-dimensional bottleneck; (ii) we propose a modular adaptation strategy that facilitates a targeted choice of adaptation layers.
>
> > **Other works in the literature that apply a low-rank constraint?**
>
> - As noted in L45-46, L79, parameter efficient finetuning methods [2, 3] that employ a low-rank constraint are becoming popular alternatives for solving downstream language tasks. In this work, we use the low-rank constraint as a means to model style shifts across domains, in addition to proposing a modular adaptation strategy.
>
> > **Importance of the low-rank constraint in the motion forecasting setup. How does the size of rank r impact performance?**
>
> - Thank you for the suggestion. To highlight the importance of a low-rank design, we have compared our low-rank MoSA design against a popular residual convolutional design (without the low-rank constraint) called Parallel Adapter (PA) [4] in Table 1 (main text). In addition, we provide a comparison with a residual feedforward adapter (without the low-rank constraint) on SDD and Level 5. For the Level 5 experiment, the batch size is 64 samples.
>
> **Table A. Analysis of choice of MoSA for SDD**
>
> | $\mathbf{N_{target}}$ (samples) | 10 | 20 | 30 |
> | :-----------: | :-----------: | :-----------: | :-----------: |
> | Conv Res. (PA) | 52.77 $\pm$ 0.85 | 47.75 $\pm$ 1.83 | 44.70 $\pm$ 1.28 |
> | FF Res. | 52.14 $\pm$ 1.61 | 46.93 $\pm$ 0.86 | 43.54 $\pm$ 0.90 |
> | MoSA (Ours) | **49.98 $\pm$ 1.05** | **45.55 $\pm$ 0.77** | **41.69 $\pm$ 0.88** |
>
> **Table B. Analysis of choice of MoSA for Level 5**
>
> | $\mathbf{N_{target}}$ (batches) | 8 | 15 | 36 |
> | ----------- | ----------- | ----------- | ----------- |
> | FF Res.| 42.37 $\pm$ 0.72 |  34.12 $\pm$ 0.51 | 30.37 $\pm$ 2.19 |
> | MoSA | **29.43 $\pm$ 0.70** | **25.96 $\pm$ 0.47** | **25.06 $\pm$ 1.26** |
>
> - We also provide performance on varying the rank r of the MoSA modules on SDD. Very low rank limits the number of style factors that can be updated leading to sub-optimal performance.  On the other hand, a high rank increases the number of trainable parameters, leading to overfitting in the low-data regime.
>
> **Table C. Analysis of the rank of MoSA for SDD**
>
> | $\mathbf{N_{target}}$ (samples) |      10 | 20 | 30 |
> | :-----------: | :-----------: | :-----------: | :-----------: |
> | Rank = 1  | 51.84 $\pm$ 1.41 | 46.68 $\pm$ 1.32 | 42.53 $\pm$ 1.19 |
> | Rank = 3  | **49.98 $\pm$ 1.05** | **45.55 $\pm$ 0.77** | **41.69 $\pm$ 0.88** |
> | Rank = 10 | 51.44 $\pm$ 0.66 | 46.48 $\pm$ 0.84 | 42.44 $\pm$ 0.72 |

---

### Official Review · Reviewer_g1Er · 2022-08-05

**Originality:** Good
**Technical Quality:** Good
**Clarity Of Presentation:** Good
**Impact:** 3

**Recommendation:**

Weak Reject: I recommend rejecting the paper, but will not argue for my recommendation if the majority of other reviewers have a different opinion.

**Summary:**

Core idea of the paper: Introduce domain adaptation techniques to the field of motion forecasting. This is demonstrated in two ways. Agent adaptation or scene adaptation. In Agent adaptation experiments, authors trained the initial model on a set of scenes on a particular agent and adapted to a different agent type from the same set of scenes. In Scene adaptation, authors trained a particular agent type on a set of scenes and demonstrated adaptation of the same agent performance on a different set of scenes. Adaptation methods is a very practical and important area of study which is proliferating in the field of NLP and vision and not so much in the field of motion forecasting. So, the core idea of the paper is quite relevant and very impactful.

**Issues:**

-

**Quality Of The Limitations Section:**

Limitations are not well addressed

**Reviewer Expertise:**

4: The reviewer is confident but not absolutely certain that the evaluation is correct

**Robotics Focus:**

Highly relevant to robotics but no hardware experiments

**Strengths And Weaknesses:**

Weaknesses

1. From figure 1, it seems like MoSA block is nothing but two linear layers without any activation function?
2. Experiments Figure 5 suggests that MoSA block outperforms fine-tuning of the full model. It is very surprising to see this conclusion as I would expect that the MoSA block tuning should either be lower or at par in performance with the fine-tuning the full model. Any intuition as to why this is the case?
3. Similarly in Figure 7 and 8: It is again very surprising to see that as we fine-tune more parts of the architecture, the overall performance is not doing better but on the contrary it is performing worse.



**Summary Of Recommendation:**

Though the work on introducing adaptation to the field of motion forecasting is new, I find the experiments to be quite in adequate and conclusions to be in effective or even counter intuitive. This suggests that he authors should re do the expeirments more carefully or provide strong supporting material or evidence along with explanation as to why full training provides overall worse performance compared to fine-tuning just the MOSA layers.

---

> ### Author Response · Authors · 2022-08-20
> **Author Response to Reviewer g1Er**
>
> Thank you for the review. Below we answer specific questions.
>
> > **MoSA block is nothing but two linear layers without any activation function**
>
> - We would like to highlight that the key design choices of MoSA are (i) the low-rank bottleneck and (ii) modular adaptation strategy. As shown in Table A and Table B below, the transfer learning results on SDD and Level 5 are worse without the low-rank constraint. Fig 7 and Fig 8 demonstrate that placing MoSA into the modules in charge of the underlying style shifts also plays a critical role in boosting the transfer performance. We hope these results bring more clarity with respect to our contributions.
>
> Table A. Analysis of the rank constraint on SDD.
> | $\mathbf{N_{target}}$ (samples) | 10 | 20 | 30 |
> | ----------- | ----------- | ----------- | ----------- |
> | No rank-constraint | 52.14 $\pm$ 1.61 | 46.93 $\pm$ 0.86 | 43.54 $\pm$ 0.90  |
> | Rank-constraint | **49.98 $\pm$ 1.05** | **45.55 $\pm$ 0.77** | **41.69 $\pm$ 0.88**  |
>
> Table B. Analysis of the rank constraint on Level 5. (Batch size = 64 samples)
> | $\mathbf{N_{target}}$ (batches) | 8 batches| 15 batches | 36 batches |
> | ----------- | ----------- | ----------- | ----------- |
> | No rank-constraint | 42.37 $\pm$ 0.72 |  34.12 $\pm$ 0.51 | 30.37 $\pm$ 2.19 |
> | Rank-constraint | **29.43 $\pm$ 0.70** | **25.96 $\pm$ 0.47** | **25.06 $\pm$ 1.26** |
>
> > **The intuition behind MoSA block outperform finetuning (in Figure 5)**
>
> - Given sufficient target data, it is reasonable to expect that the MoSA block tuning should either be lower or on par in performance with the fine-tuning of the full model. However, in this work, we are interested in model adaptation given limited samples. As explained in L33-36, fine-tuning a pre-trained model on the data collected from target domains without exploiting the inherent structure of the distributional shifts is often sample-inefficient. Our MoSA block, thanks to the low-rank constraint, greatly reduces the number of parameters to be adapted, thereby improving performance in the low-data regime as shown in Fig 5. The approach of reducing the number of trainable parameters during adaptation has shown to outperform finetuning in language and vision [1, 2] as well.
>
> > **Finetuning more parts leads to worse performance in Figure 7 and 8**
>
> - In Fig 7 and 8, we aim to demonstrate that targeted modular adaptation, i.e. add MoSA to only specific parts, further boosts model performance in low-data regime. In Fig 7, when presented with agent-motion transfer scenarios (e.g. car → trucks) that share the same physical context (see Fig. 5 of the supplementary), we propose to exploit prior knowledge and add MoSA to the agent motion encoder [A] only further reducing parameters to be learned. Empirically, we corroborate that [A] performs better than adding MoSA across all encoder modules [A+S+F], which further beats full-model finetuning that updates all model parameters.
> - In summary, we believe that the simplicity of our MoSA design combined with strong empirical results on real-world applications makes MoSA an effective adaptation technique.
>
> [1] Mahabadi et al. COMPACTER: Efficient Low-Rank Hypercomplex Adapter Layers, NeurIPS 2021
>
> [2] Chen et al. AdaptFormer: Adapting Vision Transformers for Scalable Visual Recognition, Arxiv 2022

---

### Official Review · Reviewer_SST5 · 2022-08-06

**Originality:** Good
**Technical Quality:** Very Good
**Clarity Of Presentation:** Very Good
**Impact:** 4

**Recommendation:**

Weak Accept: I recommend accepting the paper, but will not argue for my recommendation if the majority of other reviewers have a different opinion.

**Summary:**

This paper explores the problem of domain adaptation in motion forecasting. In contrast to previous methods that directly fine-tune the whole encoder, it proposes a low-rank motion style adapter that only accounts for part of the model parameters. A modular adapter strategy that disentangles the features of scene context and motion history to facilitate the fine-grained choice of adaptation layers. Experiments are conducted on Stanford Done and Intersection Drone dataset, as well as transferred to Lyft Level 5 dataset.

**Issues:**

Please answer my questions on weaknesses.

**Quality Of The Limitations Section:**

Limitations are addressed clearly

**Reviewer Expertise:**

4: The reviewer is confident but not absolutely certain that the evaluation is correct

**Robotics Focus:**

Highly relevant to robotics but no hardware experiments

**Strengths And Weaknesses:**

__Strengths__
- The problem and motivation make sense to me as the data-driven motion forecasting methods are not generalizable, and the adaptation component is useful during inference.
- The technical solutions are reasonable.
- Experiments verify the effectiveness of the method.
- The paper is easy to follow.

__Weaknesses__
- Can the authors explain the intuition that “the style shifts across forecasting domains often reside in a low-dimensional space”? I understand the statements in Line 133 that the style shifts across forecasting domains usually only exist in a small portion of latent factors. Is this consistent with “low-dimensional space”?
- Any variations or analyses on the choices of MoSA?
- Any analyses/visualization for the learned parameters/maps in MoSA? and the decoupled ones?
- The ablation studies on the Modular Motion Style Adapters are a little bit confusing. I didn’t observe conclusions in the section. It seems that for different cases, the optimal choices are different. How did the authors implement the network while inference? What’s the performance without the modular design?
- The experiments in this paper seem all transferred within the same dataset, such as pedestrian-cyclist/truck-car. However, I think this is only a small part of the applications. More comprehensive experiments shall include cross-dataset/cross-category/cross-scenes.


**Summary Of Recommendation:**

Overall I think this is a paper with reasonable motivations and solutions and deserve to be published. However, I have some concerns regarding its experiments. Hope the authors can answer these questions.

---

> ### Author Response · Authors · 2022-08-20
> **Author Response to Reviewer SST5 (Part 2/2)**
>
> > **More comprehensive experiments shall include cross-dataset/cross-category/cross-scenes.**
>
> - We would like to point out that our experiments (in Sec 4) demonstrate the effectiveness of MoSA on various cross-category and cross-scenes transfer cases albeit within the same dataset. Our focus in this work is to evaluate on refined real-world motion transfer setups where the main factor of variation across domains is restricted to motion style. In the case of cross-dataset setups, there potentially exist other factors of variations arising from differences in data collection processes, which can dominate the style variations (in magnitude).

---

> ### Author Response · Authors · 2022-08-23
> **Author Response to Reviewer SST5 (Part 1/2)**
>
> Thank you for the detailed review and suggestions! We address the raised concerns below:
>
> > **Is the description of style shifts consistent with the concept of "low-dimensional space"?**
>
> - Yes, it is a well-known result in linear algebra that if only s of d coordinates of vectors can change, then such vectors can only span a subspace of R^s that lies in R^d.  We are alluding to this fact when we state that the style shifts reside in a low-dimensional space (L44). We will reframe L44 to make it more explicit in the main text.
>
> > **Any variations or analyses on the choices of MoSA?**
>
> - We have compared our low-rank MoSA design against a popular residual convolutional design (without the low-rank constraint) called Parallel Adapter (PA) [1] in Table 1 (main text). In addition, we provide a comparison with a residual feedforward adapter (without the low-rank constraint) on SDD and Level 5. For the Level 5 experiment, the batch size is 64 samples.
>
> **Table A. Analysis of choice of MoSA on SDD**
>
> | $\mathbf{N_{target}}$ (samples)| 10 | 20 | 30 |
> | :-----------: | :-----------: | :-----------: | :-----------: |
> | Conv Res. (PA) | 52.77 $\pm$ 0.85 | 47.75 $\pm$ 1.83 | 44.70 $\pm$ 1.28 |
> | FF Res. | 52.14 $\pm$ 1.61 | 46.93 $\pm$ 0.86 | 43.54 $\pm$ 0.90 |
> | MoSA (Ours) | **49.98 $\pm$ 1.05** | **45.55 $\pm$ 0.77** | **41.69 $\pm$ 0.88** |
>
> **Table B. Analysis of choice of MoSA on Level 5**
>
> | $\mathbf{N_{target}}$ (batches) | 8 | 15 | 36 |
> | ----------- | ----------- | ----------- | ----------- |
> | FF Res.| 42.37 $\pm$ 0.72 |  34.12 $\pm$ 0.51 | 30.37 $\pm$ 2.19 |
> | MoSA | **29.43 $\pm$ 0.70** | **25.96 $\pm$ 0.47** | **25.06 $\pm$ 1.26** |
>
> - We also provide performance on varying the rank r of the MoSA modules on SDD. Very low rank limits the number of style factors that can be updated leading to sub-optimal performance.  On the other hand, a high rank increases the number of trainable parameters, leading to overfitting in the low-data regime.
>
> **Table C. Analysis of the rank of MoSA on SDD**
>
> | $\mathbf{N_{target}}$ (samples) |      10 | 20 | 30 |
> | :-----------: | :-----------: | :-----------: | :-----------: |
> | Rank = 1  | 51.84 $\pm$ 1.41 | 46.68 $\pm$ 1.32 | 42.53 $\pm$ 1.19 |
> | Rank = 3  | **49.98 $\pm$ 1.05** | **45.55 $\pm$ 0.77** | **41.69 $\pm$ 0.88** |
> | Rank = 10 | 51.44 $\pm$ 0.66 | 46.48 $\pm$ 0.84 | 42.44 $\pm$ 0.72 |
>
> > **Any visualization for the learned parameters/maps in MoSA?**
>
> - In Fig 4, we provide a heatmap showing the difference in the goal decoder output of Y-Net before adaptation and after adaptation. In addition, we provide the attention heatmaps of the last layer of vision transformer before and after model adaptation in Level 5 dataset (see attached visualization.pdf). The attention maps post adaptation are more refined and focus more on the different possible future routes and the vehicles in front.  In general, it is difficult to directly interpret the learned weights of the MoSA qualitatively. Thus, we analyze the effect of the learned weights on the intermediate representations of the forecasting model in various setups.
>
> > **Network implementation during inference for Modular Motion Style Adapters in Sec 4.4?**
>
> - **Training**:  We choose the modularized encoder architecture described in Sec 3.3 (Fig 2) as our base model.
> - **Adaptation**: We compare different adaptation configurations. For each configuration, MoSA is introduced across specific modules of the modularized encoder (see L236-238). e.g. for configuration [A]: we introduce MoSA across the agent motion encoder only.
>
> > **Clarifications regarding the ablation studies on the Modular Motion Style Adapters: It seems that the optimal choices are different for different cases.**
>
> - **Motivation**: Our main message is that if the style transfer setup is known a-priori, we can determine the specific modules of the modularized encoder that need to be adapted to provide optimal performance. For instance, in the agent motion transfer scenario, updating the agent motion module [A] should improve performance while updating the scene module [S] should deteriorate performance.
> - **Experimental Observation**: As shown in Fig. 7 and 8, different adapter configurations lead to optimal performance based on the style transfer setup. In agent motion transfer setup (Fig. 7), adapting only the agent motion module [A] performs the best, while for scene transfer setup (Fig. 8), adapting the scene module together with the fusion module [S+F] performs the best. As mentioned in L256-257, an interesting future direction of research is the automatic selection of layers to be adapted for target domains.
>
> [1] S.-A. Rebuffi et al. Efficient parametrization of multi-domain deep neural networks. CVPR, 2018.

---

### Official Review · Reviewer_PiYP · 2022-08-11

**Originality:** Good
**Technical Quality:** Good
**Clarity Of Presentation:** Good
**Impact:** 3

**Recommendation:**

Weak Accept: I recommend accepting the paper, but will not argue for my recommendation if the majority of other reviewers have a different opinion.

**Summary:**

The paper proposes a deep learning model to predict the motion trajectory of agents from the pre-trained knowledge on agents of different type. The model design is based on the premise that motion style across different type of agents include common features which can be transferred to target domain. The proposed motion style adapter uses two trainable low-rank weight matrices: one inferring the style factors that need to be matched with the target domain and the second one performs the updates. During update, the pre-trained model is frozen and the low-rank matrices are trained with a low volume samples of the target domain. The second contribution include in decoupling the style adapter into scene-specific style and agent-specific style. The experiments are conducted on 3 different data-set. The results show that the proposed model out-performs with respect to error when compared to fine-tuning, partial model fine tuning and parallel adapter design.

**Issues:**

- Please clearly state the differences from the architecture in reference 7 in the introduction.

- The authors should consider and cite the following works, especially the body of work on meta-learning:
[2] Finn, Chelsea, Pieter Abbeel, and Sergey Levine. "Model-agnostic meta-learning for fast adaptation of deep networks." International conference on machine learning. PMLR, 2017. : This work uses meta-learning to adapt to new task using small dataset.

[3] Gui, Liang-Yan, et al. "Few-shot human motion prediction via meta-learning." Proceedings of the European Conference on Computer Vision (ECCV). 2018.

- The function L($x_i$, $y_i$; $W_enc$, $W_dec$) in Equation (1) needs to be defined. Although it is intuitive in the community as Loss function, it needs to be stated for completeness of narrative as a published work. Also the authors need to state what mathematical functions it represents.

-The difference in performance between S and S+F and A and A+F in the experiments in Sec 4.4 needs to be stated in the discussion. This completes the explanation of the behavior of fusion encoder in presence and absence of two encoders.

- The size of the pre-trained data-set used in each experiment is required. Some of these detail is provided in the supplementary material. This is required to evaluate how low the volume is of target domain dataset. Also, how the performance is affected if the size of the pre-trained data-set increases or decreases and what the size of the target domain data set need to be to obtain a comparable or tolerable result. Additional experimental analysis showing the impact of the pre-trained data set size compared to the size of the target domain dataset will be insightful.

- In the experiments in Sec 4.4, it would generalize the results more if other combination of scene ids are used i.e., {1,2,3} for pre-training and 4 for adapting and others.

- Some of the details which are provided in the supplementary materials need to be moved in the main paper. This include the system architecture and how the proposed encoders are added to the existing deep learning models- the division of layers, where the adapters are applied and is there any additional hyperparameters? These details are required to implement the work without the support of supplementary.


**Quality Of The Limitations Section:**

Limitations are addressed clearly

**Reviewer Expertise:**

4: The reviewer is confident but not absolutely certain that the evaluation is correct

**Robotics Focus:**

Highly relevant to robotics but no hardware experiments

**Strengths And Weaknesses:**

The paper is clearly motivated in adapting pre-trained models to different domains. The language is easy to follow and the experimental results corroborates the claim. The experimental analysis also compares their method with other related work.
Many of the details are provided in the supplementary for the paper. Without the supplementary the architecture is not detailed and needs clarity in places. The architecture is similar to one published paper which is not cited.

**Summary Of Recommendation:**

The updated recommendation is based on the following: The issues raised in the first review were addressed in the rebuttal. However, the additional explanations as suggested need to be incorporated for publication.

---

> ### Author Response · Authors · 2022-08-20
> **Author Response to Reviewer PiYP (Part 1/2)**
>
> Thank you for the detailed review and suggestions. We address the raised concerns below:
>
> > **The architecture of decoupling encoders is similar to one of published paper (Robust and Adaptive Motion Forecasting [RA]) ……. the work is neither cited or compared.**
>
> - We would like to point out that we have cited the work (L71-72) as well as compared to RA [1] (which is baseline ET in Table 1). In fact, our implementation is built upon the public code of the RA. Please refer to the SDD_inD.zip file in our submitted code.
>
> > **Comparison to the Robust and Adaptive Motion Forecasting (RA) [1] architecture.**
>
> - **Discussion**: Both our work and RA [1] share a similar motivation, i.e. reusing the majority of pre-trained parameters for efficient adaptation. However, there exist key differences in the methodology: (1) Unlike RA, MoSA does not require access to multiple training domains with varying styles in order to perform motion adaptation (2) We propose to introduce low-rank adapters instead of finetuning existing parameters, to model domain shifts. Additionally, in our work, we demonstrate that decoupling motion style into scene-style and agent-style favors efficient adaptation, which we believe can also improve the performance of RA.
>
> - **Experiment Observation**: Table 1 shows that our MoSA outperforms RA (baseline ET) on both SDD and inD datasets. For low-shot motion style transfer on SDD (one-to-one transfer), RA proposed to adapt only the Y-Net encoder parameters (refer to Fig 8 in RA [1]). This can be viewed as a form of partial fine-tuning and more specifically encoder-only fine-tuning (ET).
>
> > **The authors should consider and cite the works on meta-learning**
>
> - Thanks for the suggestion. Indeed, meta-learning is another alternative for performing transfer learning with limited data by utilizing a transfer-oriented objective [2, 3]. In comparison, our work (1) introduces a few additional parameters responsible for modeling the domain shift while freezing the pre-trained model, (2) does not require access to a set of related tasks during training (meta-learning has this constraint to perform meta-training [2]). We will add this discussion in the related work in the updated version.
>
> > **Description of function in Equation (1)**
>
> - Let us denote the prediction of a model by $({\widehat{y}^1, ..., \widehat{y}^T})$ and the ground truth future trajectory by $({y^1, ..., y^T})$  for T time-steps; the loss function calculates the average displacement error (ADE) between the forecast and the ground truth future. Mathematically,
> $$L_{ADE} (\widehat{y}, y) = \frac{1}{T} \sum_{t=1}^T ||\widehat{y}^t - y^t||_2.$$
> We utilize the above objective for training the Vision Transformer on the Level 5 dataset. For SDD/inD experiments, we sample 20 forecasts from the model using the sampled strategy described in [4] and then evaluate the ADE of the forecast closest to the ground truth. We will add this description to the text.
>
> > **What is the purpose of the fusion encoder?**
>
> - The fusion encoder fuses the representations of the physical scene and the agent past motion. Once we obtain the embedding of these modalities using the two low-level encoders (see Fig. 2), following recent works [5, 6], we introduce the fusion encoder that combines the information from all modalities to generate a representation of the environment.
>
> > **Why is there a difference in performance between S and S+F and A and A+F in the experiments in Sec 4.4?**
>
> - Our main message is that if the style transfer setup is known a-priori, we can determine the specific modules of the modularized encoder that need to be adapted to provide optimal performance.
> - **Training**:  We choose the modularized encoder architecture described in Sec 3.3 (Fig 2) as our base model.
> - **Adaptation**: We compare different adaptation configurations. For each configuration, MoSA is introduced across specific modules of the modularized encoder (see L236-238). e.g. for configuration [A]: we introduce MoSA across the agent motion encoder only.
> - **Experimental Observation**:  In the scene transfer experiment, [A] performs worse than [A+F] as MoSA applied to [A] cannot modify the physical context representation. While, in [A+F] configuration, the adapters of the fusion encoder can modify the physical scene embedding.
> - Following similar reasoning, in the car → truck (agent motion transfer) experiment, [S] performs worse than [S+F]. Moreover, [A] performs that best as we only need to update the agent-motion module: the physical scene context shared by cars and trucks is identical (see Fig. 5 of the supplementary). Thus, we show that different configurations result in different performances dependent on the style transfer setup.

---

> ### Author Response · Authors · 2022-08-20
> **Author Response to Reviewer PiYP (Part 2/2)**
>
>
> > **Additional experimental analysis showing the impact of the pre-trained data set size compared to the target domain dataset size will be insightful.**
>
> - Thanks for your suggestion. As suggested, we vary the size of the pre-trained dataset while keeping the target domain dataset size fixed. We report the performance of MoSA on different sizes of Level 5 pre-training in Fig. A of the attached plots.pdf.  We make the following observations: (1) The bigger the size of the pre-trained dataset relative to the target domain dataset, the better the adaptation performance. (2) The adaptation performance starts to saturate after a certain point.
>
> > **It would generalize the results of modularization more if other combinations of scene ids are used i.e., {1,2,3} for pre-training and 4 for adapting and others.**
>
> - Thanks for your suggestion. Indeed, different scene combinations can help to corroborate the efficacy of our modular adaptation strategy. As shown in Fig. B of the attached plots.pdf, we provide the results of our strategy for a different setup of scene generalization. We observe the same trend as Fig. 8: (1) updating the scene module [S] is more effective than updating all the modules [S+A+F]. (2) Adapting of agent motion module [A] deteriorates performance.
>
> > **Some of the details in the supplementary materials need to be moved to the main paper.**
>
> - As suggested, we will move the system architecture details and hyperparameter information to the main text to enable the reader to reproduce results without referring to supplementary.
>
> [1] Liu, Yuejiang, et al. "Towards Robust and Adaptive Motion Forecasting: A Causal Representation Perspective." Proceedings of the IEEE/CVF Conference on Computer Vision and Pattern Recognition. 2022.
>
> [2] Finn, Chelsea, Pieter Abbeel, and Sergey Levine. "Model-agnostic meta-learning for fast adaptation of deep networks." International conference on machine learning. PMLR, 2017.
>
> [3] Gui, Liang-Yan, et al. "Few-shot human motion prediction via meta-learning." Proceedings of the European Conference on Computer Vision (ECCV). 2018.
>
> [4] Karttikeya Mangalam et al. “From Goals, Waypoints & Paths To Long Term Human Trajectory Forecasting.” ICCV 2021
>
> [5] Sarthak Mittal et al. “Is a Modular Architecture Enough?”, Arxiv 2022
>
> [6] Nigamaa Nayakanti et al. “Wayformer: Motion Forecasting via Simple & Efficient Attention Networks”, Arxiv 2022

---

### Meta-Review · Area_Chair_ZSPT · 2022-08-09

**Recommendation:** Accept (Poster)
**Confidence:** 4

**Metareview:**

Summary: This paper addresses domain adaptation for motion forecasting. Unlike earlier methods, it provides a low-rank motion style adaptor that only accounts for a portion of the model parameters. They introduce domain adaptability to motion prediction. There are two ways to demonstrate this. Agent/scene adaptation. In Agent adaptation experiments, authors trained a model on a series of scenes involving one agent type and then adapted it to a different agent type. In  scene adaptation  an agent type  is trained on a set of scenes and demonstrated adaptation to a different set. A further decomposition is investigated in which scene and agent information are modularized and distinct weights are based on labeled data (agent, scene, or both have changed). Comparisons to competing approaches on three datasets demonstrate that the suggested strategy achieves improved standard error metric performance.

Strengths
1.	Data-driven motion forecasting approaches cannot be easily generalized and the adaption component presented is useful for inference.
2.	The experiments demonstrate the method's efficacy.
3.	 Motion forecasting models as presented in this paper show that they need to be changed to new conditions, and that low-data adaption techniques are crucial.
4.	It makes sense to train a residual model that is limited to have a low rank in order to modify the model while maximizing a small number of parameters which is important in the low-data regime).
5.	The method presented are well-explained.

Weaknesses
1.	More explanation is needed to explain why style shifts across forecasting domains frequently reside in a low-dimensional space and if the definition presented is consistent with the concept of "low-dimensional space"?
2.	More clarity is needed about the Modular Motion Style Adapters and what conclusions were made.
3. The paper does not show the implementation of the network with inference and the performance result without the modular design.
4.	The experiments in this paper seem all transferred within the same dataset, such as pedestrian-cyclist/truck-car.  More comprehensive experiments shall include cross-dataset/cross-category/cross-scenes.
5.	The MOSA block as presented reads like the two linear layers without any activation function. The contribution of this MOSA needs to be better explained.
6.	the method makes sense for the low-data regime, It is not clear how novel the idea of adaptation by training a residual over a frozen model is.
7.	The low rank constraint on the residual layer is an interesting addition. The authors should investigate similar work on meta-learning/finetuning that apply a similar constraint.

Update:
We thank the authors for making relevant revisions to the paper and responding to the concerns of the reviewers.


**Best Paper Nomination:**

No

---

> ### Author Response · Authors · 2022-08-27
> **Author Response to Area Chair ZSPT**
>
> Thank you very much for the detailed meta-review. Please find below a summary of our answers to the raised concerns. More details can be found in our responses to each reviewer separately.
>
> > **The low-rank constraint on the residual layer is an interesting addition. More explanation is needed to explain why style shifts across forecasting domains frequently reside in a low-dimensional space.**
>
> - We highlight the importance of a low-rank design by comparing low-rank MoSA against a popular residual convolutional design (without the low-rank constraint) called Parallel Adapter (PA) [1] in Table 1. In addition, we provided a comparison to a residual feedforward adapter (without the low-rank constraint) on SDD and Level 5 (see Table A and Table B in the response to reviewer SST5) and showed that MoSA performed superior to both designs.
>
> - We provide performance on varying the rank r of the MoSA modules on SDD (see Table C in the response to reviewer SST5). Very low rank limits the number of style factors that can be updated leading to sub-optimal performance. On the other hand, a high rank increases the number of trainable parameters, leading to overfitting in the low-data regime.
>
> > **The MOSA block as presented reads like the two linear layers without any activation function. The contribution of this MOSA needs to be better explained.**
>
> We would like to highlight that the key design choices of MoSA are (i) the low-rank bottleneck and (ii) the modular adaptation strategy. Table A and Table B (in the response to reviewer SST5) demonstrate the importance of the low-rank constraint, while Fig 7 and Fig 8 demonstrate highlight the effectiveness of the modular adaptation strategy.
>
> > **Is the presented definition of style shifts consistent with the concept of "low-dimensional space"?**
>
> Yes, it is consistent, and we have clarified this in the response to reviewer SST5.
>
> > **How did the authors implement the network while inference for Modular Motion Style Adapters in Sec 4.4?**
>
> We have clarified the implementation strategy in the response to reviewer SST5
>
> > **More clarity is needed about the Modular Motion Style Adapters and what conclusions were made.**
>
> - Our main message is that if the style transfer setup is known a-priori, we can determine the specific modules of the modularized encoder that need to be adapted to provide optimal performance.
> - As shown in Fig. 7 and 8, in agent motion transfer setup, adapting only the agent motion module [A] performs the best (in comparison to adapting the entire encoder), while for scene transfer setup (Fig. 8), adapting the scene module together with the fusion module [S+F] performs the best. Thus, we conclude that our modular adaptation strategy helps to boost the adaptation performance (see L10-12).
>
> > **The authors should investigate similar work on meta-learning/finetuning that applies a similar constraint.**
>
> We have compared the proposed works in the responses to reviewer PiYP and reviewer Qhox. We will add the discussion in the related work in the updated version.
>
> > **The method makes sense for the low-data regime. It is not clear how novel the idea of adaptation by training a residual over a frozen model is.**
>
> Unlike the conventional residual block densely injected into the encoder, our method emphasizes on the importance of the (i) low-rank bottleneck and (ii) sparse modular strategy. Please refer to our more detailed response to reviewer Qhox.
>
> > **The experiments in this paper seem all transferred within the same dataset, such as pedestrian-cyclist/truck-car. More comprehensive experiments shall include cross-dataset/cross-category/cross-scenes.**
>
> We would like to point out that our experiments (in Sec 4) demonstrate the effectiveness of MoSA on various cross-category and cross-scenes transfer cases. We have addressed the cross-dataset concern in the response to reviewer SST5.
>
> [1] S.-A. Rebuffi, H. Bilen, and A. Vedaldi. Efficient parametrization of multi-domain deep neural networks. In Proceedings of the IEEE Conference on Computer Vision and Pattern Recognition, 2018.